# Informative Path Planning Using Physics-Informed Gaussian Processes for Aerial Mapping of 5G Networks

**DOI:** 10.3390/s24237601

**Published:** 2024-11-28

**Authors:** Jonas F. Gruner , Jan Graßhoff , Carlos Castelar Wembers , Kilian Schweppe , Georg Schildbach , Philipp Rostalski 

**Affiliations:** 1Institute of Electrical Engineering in Medicine, Universität zu Lübeck, 23562 Lübeck, Germany; jonas.gruner@student.uni-luebeck.de (J.F.G.); carlos.castelar@uni-luebeck.de (C.C.W.); georg.schildbach@uni-luebeck.de (G.S.); philipp.rostalski@uni-luebeck.de (P.R.); 2Fraunhofer IMTE, Fraunhofer Research Institution for Individualized and Cell-Based Medical Engineering, 23562 Lübeck, Germany

**Keywords:** unmanned aerial vehicle, drone, path planning, mapping, mobile network, 5G, Bayes methods, Gaussian processes

## Abstract

The advent of 5G technology has facilitated the adoption of private cellular networks in industrial settings. Ensuring reliable coverage while maintaining certain requirements at its boundaries is crucial for successful deployment yet challenging without extensive measurements. In this article, we propose the leveraging of unmanned aerial vehicles (UAVs) and Gaussian processes (GPs) to reduce the complexity of this task. Physics-informed mean functions, including a detailed ray-tracing simulation, are integrated into the GP models to enhance the extrapolation performance of the GP prediction. As a central element of the GP prediction, a quantitative evaluation of different mean functions is conducted. The most promising candidates are then integrated into an informative path-planning algorithm tasked with performing an efficient UAV-based cellular network mapping. The algorithm combines the physics-informed GP models with Bayesian optimization and is developed and tested in a hardware-in-the-loop simulation. The quantitative evaluation of the mean functions and the informative path-planning simulation are based on real-world measurements of the 5G reference signal received power (RSRP) in a cellular 5G-SA campus network at the Port of Lübeck, Germany. These measurements serve as ground truth for both evaluations. The evaluation results demonstrate that using an appropriate mean function can result in an enhanced prediction accuracy of the GP model and provide a suitable basis for informative path planning. The subsequent informative path-planning simulation experiments highlight these findings. For a fixed maximum travel distance, a path is iteratively computed, reducing the flight distance by up to 98% while maintaining an average root-mean-square error of less than 6 dBm when compared to the measurement trials.

## 1. Introduction

The emergence of the new mobile standard, 5G, represents a revolutionary step in terms of data rates, latency times, massive connectivity, network reliability, and energy efficiency. High data rates and low latency make this technology especially useful in modern industry. Private local 5G networks enable secure and independent access to this novel network standard and the benefits it may provide. However, the full potential can only be reached if the network reception is reliable over an entire industrial site [1]. The 5G standard allows for higher frequencies compared to previous standards like LTE and UMTS (3G & 4G), making it more susceptible to path loss and blockage effects [1]. Thus, detailed planning, including ray-tracing simulations of antenna deployment, can be very beneficial [2]. However, the simulation is only as good as the underlying geospatial data and the propagation model. The results may, therefore, deviate from the actual physical network due to unmodeled structures and obstacles or other sources of error [2]. Monitoring and structured measurement of the physical network using an unmanned aerial vehicle (UAV) can provide insights into unforeseen weak spots that can be avoided by changing the antenna orientation or adding new antennas to the network if necessary. This helps to improve its performance and reliability. Furthermore, compliance with legal limits on signal strength, especially at the boundaries of the network site, can also be confirmed with such measurements.

To avoid missing important locations, i.e., areas with insufficient coverage or areas exceeding legal limits, the UAV’s flight path usually follows a dense grid while flying at low speed. However, such a flight path contradicts the battery-power limitation of common multicopter configurations for UAVs. To the best of our knowledge, previous structured cellular network coverage measurements conducted using a UAV have been limited to small one- or two-dimensional measurement areas (cf., [3]). For instance, Raffelsberger et al. [4] and Sundqvist [5] presented only one-dimensional measurements in a 4G LTE network, while Platzgummer et al. [6] conducted two-dimensional measurements in a 5G network. To average out measurement noise, they stopped at waypoints along the measurement trajectory and considered only the average measurement at those waypoints, applying natural neighbor interpolation to the cluster centroids and their means to create a two-dimensional map.

Di Taranto et al. [7] and Anh et al. [8] showed that Gaussian Processes (GPs) are beneficial for predicting signal strengths in mobile networks. GPs are nonparametric, kernel-based probabilistic models that can incorporate prior information about an unknown function by means of their mean and covariance function [9]. The common simplification of the GP mean to a zero function is not well suited for extrapolation because the posterior mean converges to zero in areas with fewer measurements. Thus, Di Taranto et al. [7] and Anh et al. [8] incorporated path loss and fading effects into GP mean and covariance, respectively. However, to the best of our knowledge, in the case of UAV mapping and path planning, it has not yet been investigated how the mean function influences the GP prediction performance.

In this work, we present novel and comprehensive three-dimensional 5G signal strength measurements in volumes of up to $1.7\times 10^{4}$ $m^{3}$ within a cellular campus network at the largest German harbor on the Baltic Sea, the Skandinavienkai of the Port of Lübeck, Germany. We show that GP regression is suitable for predicting missing information and identifying regions that require further examination. Furthermore, GPs inherently provide uncertainty estimates with any prediction. This approach also enables larger three-dimensional measurement areas to be surveyed despite limited flight time. Instead of stopping at waypoints to average out measurement noise (cf., [6]), the GP models the measurement noise as Gaussian white noise.

The first main contribution of this article is the extension of the idea of Di Taranto et al. [7] by analyzing the influence of different physics-informed mean priors of varying complexity, including a full ray-tracing simulation, as the GP mean function. In order to address the sequential nature of data collection in a realistic UAV mapping application, we evaluate the performance using a more natural (and challenging) split between training and testing data points. We purposefully use the data points from the measurement data in the order they were collected instead of random points from the entire training set.

The second main contribution is the application of a GP model with physics-informed mean functions as a basis for informative path planning (IPP) of the UAV flight path. The IPP problem, also known as robotic information gathering (cf., [10]), is concerned with planning the trajectory of a robot, i.e., a UAV, with a measurement unit to collect informative measurements for efficient mapping of a search space with as few measurements or as short a path length as possible. Available approaches include continuous IPP using Bayesian optimization [11,12], evolutionary strategies [13], and iterative candidate selection approaches [10,14], which differ in their sampling and selection strategies. The algorithms for a continuous IPP proposed by Marchant, Morere, and Ramos using continuous splines with numerous free parameters [11] or partially observable Markov decision processes (POMDPs) [12] are too computationally complex for online path planning onboard a UAV. In the candidate selection method proposed by Yang et al. [10,14], a utility function based on mutual information is maximized. However, a fixed target is approached, making it unfit for a mapping task. For this reason, we choose to draw random candidate points across the search area and select the candidate with the highest expected reward. Similarly, the expected entropy of each point was used as a reward by Chen et al. [10] for their active sampling strategy, but the measurements collected along the path to that point were omitted. We present a novel combined algorithm that defines a continuous informative path within a three-dimensional search area using a fixed maximum total length of all segments. Different configurations of the approach are assessed in a real-time hardware-in-the-loop simulation using the M300 UAV platform by DJI (Shenzhen, China).

The third central contribution of this paper is the development of a UAV-based 5G measurement system that accurately performs a three-dimensional mapping of a 5G network site within the premises of the Port of Lübeck using physics-informed GP-based data processing.

Following this introduction, Section 2 presents the experimental measurement trials that later serve as ground truth. In Section 3, the Gaussian process model for signal strength prediction and the investigated mean functions are presented. Section 4 includes the evaluation setup that is used to compare the performance of the mean functions in different training and test split settings. The results of the evaluation are presented and discussed in Section 4.1 and Section 4.2. An IPP simulation using the evaluated models is introduced in Section 5. The evaluation setup and the results of informative path planning are presented in Section 5.1 and Section 5.2. Finally, the paper concludes with a discussion of future work in Section 6.

## 2. Experimental Measurements

Experimental measurement trials carried out at the Port of Lübeck, Germany, serve as ground truth. They are used to construct the training and test sets for the evaluation and the informative path-planning simulation.

The 5G measurement unit consists of an LPPM[X]-36-55-[VAR] 4x4 MIMO antenna manufactured by Panorama Antennas Ltd. (London, UK) and an RM500Q 5G modem manufactured by Quectel (Shanghai, China). The modem is attached to a Raspberry Pi Compute Module 4 (CM4) on a Dual Gigabit Ethernet 5G/4G Base Board manufactured by Waveshare (Shenzhen, China). The significance of the combined baseboard lies in the fact that a USB connection between the single-board computer and the modem may result in significant interference with the global navigation satellite system (GNSS) of the UAV (cf., [15]).

For a safe and stable flight, the UAV has to have a sufficient payload to carry the measurement unit with a good margin. The Matrice 300 (M300) quadcopter by DJI (Shenzhen, China) was chosen as a platform due to its 2.7 kg payload capacity. It also offers an abundance of obstacle avoidance sensors and an onboard software development kit (OSDK) to control the UAV and collect telemetry data. Onboard communication with the M300 UAV platform is achieved using a Manifold 2-C single-board computer by DJI (Shenzhen, China) mounted on top of the UAV. The measurement unit is mounted below the UAV body and connected to the computer via Ethernet cable, again avoiding a USB connection. The mounted measurement setup is shown in Figure 1.

Data acquisition and synchronization are achieved using the Robotic Operating System (ROS) publisher and subscriber framework. The CM4 queries signal parameters via AT commands using a serial connection to the modem. The data are then published in an ROS topic. Similarly, the manifold uses the OSDK to publish GNSS information on an ROS topic. A data logger subscribes to both topics using a rate of 1 Hz for the signal strength publisher.

Flight planning, as depicted in Figure 2, is performed using UgCS ground control station software by SPH Engineering (Riga, Latvia). Detailed LiDAR point clouds and multispectral imagery are acquired with a Zenmuse L1 LiDAR by DJI (Shenzhen, China) mounted on the M300 UAV platform. The geographic information is used to create no-fly zones (indicated by red transparent areas) around higher obstacles like light poles. This is an additional safety layer to the onboard obstacle avoidance mechanism of the M300 UAV platform. A line pattern across two volumetric survey areas covering four different heights at ground areas of about 1 ha and 2.5 ha was created with a horizontal distance between two lines of 20 m and a vertical distance of 2.5 m. A flight speed of 5 m/s is chosen, yielding a measurement every five meters and totals of 1500 and 2949 data points, respectively. The flight routes are split into multiple parts because a single flight cannot exceed 25 minutes due to battery limitations. Each part covers two of the four height layers.

The resulting measurements of the received reference signal strength (RSRP) are shown in Figure 3. The simulated antenna pattern is also plotted to highlight how the measurements coincide with the direction of the antenna’s main lobe.

The measurements for the cuboid on the left side of Figure 3 were performed twice with the same flight route. The root-mean-square error of the one trial was compared to the nearest neighbor interpolation of the other trial. The error between the interpolation of the first trial and that of the second trial was 10.23 dBm.

## 3. Gaussian Processes for Mobile Networks

In this work, the predictive mean and variance function are given by a GP regression model. The model presents a Gaussian distribution at each candidate location (x). Observations (y=f(x)+ϵ) are modeled with Gaussian white noise ($\epsilon ∼N(0,\;\sigma_{n}^{2})$). The $\sigma_{n}^{2}$ parameter is often called measurement noise and is one of the hyperparameters (θ) of the GP. A Gaussian process prior specified by a mean function (m(x)) and a covariance function ($k(x,x^{\prime })$) is used to model the unknown function (f(x)):(1)$$f\left(x\right)∼\mathrm{GP}(m\left(x\right),k\left(x,x^{\prime }\right)).$$

The predictive posterior distribution of the signal strength at a new location ($x^{*}$), given the current dataset ($X_{1:n}=\left\{x_{1:n},y_{1:n}\right\}$), can be computed as
(2a)$$\begin{array}{cc} {\bar{f}}^{*} & =m\left(x^{*}\right)+k\left(x^{*},x_{i}\right)K^{-1}\left(y-m(x_{i})\right), \end{array}$$
(2b)$$\begin{array}{cc} V^{*} & =k\left(x^{*},x^{*}\right)-k{\left(x^{*},x_{i}\right)}^{⊤}K^{-1}k\left(x^{*},x_{i}\right). \end{array}$$
where *K* denotes the covariance matrix with entries of ${\left[K\right]}_{ij}=k\left(x_{i},x_{j}\right)+\sigma_{n}^{2}\delta_{ij}$, where $\delta_{ij}=1$ for i=j and zero otherwise. $k(x^{*},x_{i})$ results in the n×1 vector of cross-covariances between $x^{*}$ and the training inputs ($x_{i}\in X_{1:n}$). The Matérn kernel for ν=1/2 defined as
(3)$$k_{M}^{1/2}\left(x,x^{\prime }\right)=\sigma^{2}\exp\left(-\frac{|x-x^{\prime }|}{\ell }\right),$$
is used as a covariance function. Here, σ represents the signal variance, *ℓ* denotes the length scale, and x and $x^{\prime }$ are input vectors. The 1/2-Matérn kernel was chosen because it performs well compared to other common kernels, i.e., the squared exponential kernel used by Di Taranto et al. [7]. All evaluation calculations (see Section 4) were also performed for a squared exponential kernel, as well as 3/2-Matérn and 5/2-Matérn kernels. In order to highlight the influence of the mean functions, the results of the other kernels are not shown for the sake of clarity and brevity.

The hyperparameters of the kernel are bounded to physically reasonable values for the mobile network setting. The kernel variance (σ) is bounded between 1 and 100, and the length scale (*ℓ*) is bounded between 0 and 250. Furthermore, the noise variance ($\sigma_{n}$) is kept fixed on an initial value computed using the measurement data. This avoids very high trained hyperparameter values, especially for the zero-mean GP, which may lead to numerical instability. High values of kernel hyperparameters can occur if the training data are not very informative.

The GPs are implemented in Python 3.7 using the GPFlow package [16] in combination with TensorFlow [17].

### 3.1. GP Mean Function

The mean function ($m(x^{*})$) is commonly simplified to the zero function ($m_{\mathrm{ZERO}}\left(x\right)$). However, this makes the posterior mean converge to zero in areas with no or few measurements. In the case of mobile network measurements, prior information can be incorporated into the mean function. Shadowing effects can also be represented in the kernel function. To this end, Di Taranto et al. used a path-loss equation as a mean function and a spatial auto-covariance function [7]. We adapt the idea for the kernel and compare different mean functions. The mean functions vary in their complexity and the number of hyperparameters that have to be learned from training data.

#### 3.1.1. Simulation-Based Mean

Knowledge about cell tower locations and antenna characteristics allows for the use of a detailed ray-tracing simulation as part of the mean function. The simulation is performed using the MATLAB 2023a Antenna Toolbox and a detailed 3D-OpenStreetMap of buildings in the mapping area. Cell tower locations and antenna configurations are included based on the preliminary setup of the 5G network at the Skandinavienkai of the Port of Lübeck, Germany. The point-cloud-based elevation information that was already used for the planning of the measurement flights described in Section 2 was translated into a 3D-OpenStreetMap using ArcGIS Pro by Esri (Redlands, CA, USA). The process and the resulting simulation for one example cell are visualized in Figure 4.

The ray-tracing simulation is computed for one individual cell on a dense spatial grid, then interpolated linearly using nearest neighbors for use as a continuous mean function:(4)$$m_{\mathrm{SIM}}\left(x\right)=m_{0}+P_{\mathrm{SIM}}\left(x\right),$$
where $m_{0}$ is an offset parameter used to tune the magnitude of the simulation to the measurements. It is learned as a hyperparameter based on training data.

#### 3.1.2. Distance-Based Mean

A simple mean function similar to the path-loss expressions presented by Zeng et al. [3] can be constructed by using the Euclidean distance to the cell tower ($d_{3D}$) and the known altitudes of the base station ($h_{\mathrm{BS}}$) and the UAV ($h_{\mathrm{UAV}}$):(5)$$m_{\mathrm{CD}}\left(x\right)=m_{0}+\alpha \cdot d_{3D}\left(x\right)+\beta \cdot \log_{10}\left(\frac{h_{\mathrm{UAV}}\left(x\right)}{h_{\mathrm{BS}}}\right).$$

The distance-based mean now has three hyperparameters( $m_{0}$, α, and β) that need to be learned from data.

#### 3.1.3. 3GPP Mean

Another promising basis for a prior mean is a formula collection based on measurement data from the 3rd Generation Partnership Project (3GPP) [18,19]. Based on parameters such as the terrestrial distance to the nearest cell tower ($d_{2D}$) and the altitudes of both the UAV ($h_{\mathrm{UAV}}$) and the base station ($h_{\mathrm{BS}}$), path losses (PL) and probabilities (Pr) for LOS/non-LOS (NLOS) connections to the base station are calculated. The received signal power ($P_{3\mathrm{GPP}}$) can then be estimated as follows:(6)$$\begin{array}{c} \begin{array}{cc} P_{3\mathrm{GPP}}\left(x\right)= & Pr_{\mathrm{LOS}}\left(x\right)\cdot PL_{\mathrm{LOS}}\left(x\right)\\ & -\left(1-Pr_{\mathrm{NLOS}}\left(x\right)\right)\cdot PL_{\mathrm{NLOS}}\left(x\right). \end{array} \end{array}$$

The 3GPP provides tabulated coefficients to calculate $Pr_{\mathrm{LOS}/\mathrm{NLOS}}\left(x\right)$ and $PL_{\mathrm{LOS}/\mathrm{NLOS}}\left(x\right)$. The coefficients are dependent on the environments where both the cell tower and the receiver are located. A distinction is made between urban (“Urban Micro, UMi”), suburban (“Urban Macro, UMa”), and rural (“Rural Macro, RMa”) environments.

Based on the assumptions made by the 3GPP, a local 5G campus network at an industrial site coincides most closely with the UMi environment. Using the corresponding coefficients, the resulting mean function, i.e.,
(7)$$m_{3\mathrm{GPP}-\mathrm{UMi}}\left(x\right)=m_{0}+P_{3\mathrm{GPP}-\mathrm{UMi}}\left(x\right),$$
also uses $m_{0}$ as an offset parameter that is learned as a hyperparameter based on training data.

Alternatively, the coefficients can also be learned as additional hyperparameters. The resulting mean function ($m_{3\mathrm{GPP}}$) has a total of six free hyperparameters. Note that the mean function only covers the LOS case, while the NLOS case of (Equation 6) is disregarded for simplification. For the sake of clarity, the non-linear logarithmic expressions for the path loss ($PL_{\mathrm{LOS}}$) and the probability ($Pr_{\mathrm{LOS}}$) are not written out. The interested reader is referred to [18,20] for further details.

## 4. Mean Function Evaluation

The performance of the GP prediction using different mean functions is evaluated for one cell. Different settings for of the training points are used:(a)Training points from a different height than the test points;(b)Training points from a small cluster within an area distant from the base station;(c)Training points from a large measurement set far away from the test set;(d)Training points from the entire measurement set, selected using a random split.

For settings (a)–(c), an increasing number of training points is sampled from the training sets. The test points are kept fixed to all points that are not part of the training set. In setting (d), an increasing number of training points is sampled from the entire measurement set, while the remaining points are used as test points. The different settings are visualized in Figure 5. The training points are sampled in the same order as they appear in the measurement set. However, the index of the first element to be used is chosen at random in every iteration. In this way, samples are locally dependent on purpose, as this mimics how a UAV sensor unit would collect the data.

For the evaluation, the root-mean-square error between the predicted value of the utilized GP and the measurement from the test set is calculated for each test-set size and in each setting. The different sizes of the training sets limit the maximum number of training points that can be used. For each setting, the performance achieved using only the interpolated ray-tracing simulation-based mean function (Equation 4) as a predictor is also evaluated (“onlySIM”). For the $m_{0}$ parameter, the same value that the corresponding GP computed as a hyperparameter is used. This allows for direct investigation of the influence of the GP.

### 4.1. Evaluation Results

The evaluation results for the different settings are shown in Figure 6. The root-mean-square error (RMSE) between the measurement from the test set and the predicted value of the GP with the indicated mean function ($m_{3\mathrm{GPP}-\mathrm{UMi}}$, $m_{\mathrm{CD}}$, $m_{3\mathrm{GPP}}$, $m_{\mathrm{SIM}}$, $m_{\mathrm{ZERO}}$) or the simulation without a GP (“onlySIM”) is plotted in different colors using logarithmic scaling.

It can be seen that in all settings, the RMSE of the simulation-based mean function ($m_{\mathrm{SIM}}$) and the simulation without a GP (“onlySIM”) is mostly independent of the number of training points. For settings (a) and (b) (see Figure 5) “onlySIM” cannot be distinguished from the GP-based $m_{\mathrm{SIM}}$. However, in setting (c), the RMSE for “onlySIM” is even lower than that of the GP variant, starting at around 350 training points. For setting (d), it is the other way around, starting at roughly 250 training points. This indicates that the GP can improve the prediction if the prediction location is not too far away from the training points. However, it is also evident that in settings (b) and (d), both simulation-based configurations are outperformed by the other GPs for larger numbers of training points.

Both $m_{\mathrm{ZERO}}$ and $m_{3\mathrm{GPP}}$ are unable to significantly improve within settings (a) and (c). For settings (b) and (d), the improvement is correlated with the number of training points.

$m_{\mathrm{CD}}$m on the other hand, shows a visible improvement with the first few training points. In setting (d), $m_{\mathrm{CD}}$ shows the lowest RMSE for most runs above 250 training points.

The pre-parameterized $m_{3\mathrm{GPP}-\mathrm{UMi}}$ mean function starts with the lowest RMSE when using fewer than 15 training points, except for the simulation-based configurations. However, with more training points, it is outperformed by the $m_{\mathrm{CD}}$ mean.

### 4.2. Evaluation Discussion

The results highlight three main findings. First, a mean function that incorporates prior knowledge on the data can be beneficial, especially if smaller numbers of locally dependent training points are expected or if the training points come from a different, possibly less informative area than the test points. If this is not the case, the GP prediction depends solely on the kernel. This is indicated by the corresponding equal or better performance of the zero mean.

As a second finding, it can be deduced that the performance of the GP mean functions is influenced by their degrees of freedom, i.e., the number of hyperparameters. Only one hyperparameter, as found in the simulation-based mean and the 3GPP-based mean, provides a very consistent RMSE compared to the test set, mostly independent of the number of training points. This is useful for very small training sets but hinders the learning capabilities of the GP for larger training sets; as evidenced by the comparison with a zero-mean GP or the sole simulation without a GP. In contrast to one hyperparameter, the six hyperparameters of the 3GPP-based mean without tabulated values ($m_{3\mathrm{GPP}}$) require a lot of training points to reach a similar RMSE compared to the simulation and even the non-informative zero-mean, which have a similar or even lower RMSE. The three hyperparameters in the distance-based mean ($m_{\mathrm{CD}}$) may provide a good middle ground for the prediction performance.

Finally, the prediction performance of the simulation is superior to the other GP means on training set configurations where the training data are not very informative in comparison to the test set. The GP is unable to learn certain trends and correlations in the data if the information is not available in the training data. For the application of cellular measurements, $m_{\mathrm{CD}}$ may provide a good compromise between the very consistent but barely improving simulation-based mean and the zero mean, which is highly reliant on training data. For the informative path-planning simulation investigated in the remainder of this paper, the distance-based and simulation-based mean functions are used and compared with the zero mean.

## 5. Informative Path Planning for Mobile Network Mapping

The previous sections presented how physics-informed GPs can help create a reliable three-dimensional map of a cellular network site. The following elaborates on these results by defining a path for the UAV to collect measurements and create a reliable map with a fixed travel budget. The path should minimize the root-mean-square error between the ground-truth measurement data and the prediction of our physics-informed GP model. It should be adjusted as new measurements become available that allow for updating of the GP posterior. Such a path-planning problem is also known as informative path planning (IPP).

The probabilistic nature of the GP-based model encourages the adoption of a Bayesian optimization (BO) approach, as detailed previously by Brochu et al. [21]. BO is a highly efficient optimization technique for functions that demand expensive evaluations. However, BO, as a global optimizer, suggests a single point ($x^{*}$) to evaluate next to find the optimum function value. To iteratively define the UAV’s trajectory for an IPP mapping task, a candidate selection approach similar to those proposed by Yang et al. [14] and Chen et al. [10] is used. For a specified number of random points ($p_{i}=x_{i}^{*}$) within the mapping area, a BO utility function considering both the mean and uncertainty of the GP is evaluated. Additionally, the Euclidean distance from the UAV’s current location to each candidate point, denoted as $d_{i}$, is computed. This approach is visualized in Figure 7. It is important to note that unlike the depicted points and the approaches proposed by Yang et al. and Chen et al., the candidate points are chosen in a three-dimensional search space instead of on a two-dimensional plane.

The BO utility value is calculated using an upper confidence bound (UCB, $a_{\mathrm{UCB}}$) modified to identify regions with a high gradient (cf., [12]):(8)$$\nabla a_{\mathrm{UCB}}\left[p_{i}\right]=\left|\nabla \mu_{\mathrm{GP}}\left(p_{i}\right)\right|+\kappa \sigma_{\mathrm{GP}}\left(p_{i}\right).$$
where $\nabla \mu_{\mathrm{GP}}$ and $\sigma_{\mathrm{GP}}$ represent the gradient of the predictive mean and the variance of the GP, respectively. The κ parameter balances exploration and exploitation. In order to also assess data points that would be collected on the way to the candidate point, the BO utility is summed up along a straight line to the candidate point using a fixed interval that matches the flight speed (*v*) of the UAV:(9)$$r\left(p_{i};M_{i}\right)=\frac{1}{M_{i}}\sum_{m=0}^{M_{i}}\nabla a_{\mathrm{UCB}}\left[l\left(p_{i},m,M_{i}\right)\right].$$
with $M_{i}=d_{i}/v$ and
(10)$$l\left(p_{i},m,M_{i}\right)=p_{0}+\frac{m}{M_{i}}\left(p_{i}-p_{0}\right),$$
where $p_{0}$ denotes the present location of the UAV. Both the BO reward and the distance to the candidate point are individually ranked, with the highest BO reward (*r*) and the smallest distance (*d*) receiving the highest rank. This normalization facilitates a direct trade-off achieved by summing the ranks with a tuning factor (α), which provides flexibility to tune the path length to the next waypoint. Subsequently, the UAV navigates to the candidate point with the highest summed rank, acquiring new measurements along the way. In the subsequent iterations, the GP model is updated with the new data, and fresh candidate points are selected and evaluated. This iterative process continues until the summed-up path lengths reach a travel budget or a predefined maximum number of iterations is surpassed. Iterative path planning is summarized in Algorithm 1.  

**Algorithm 1:** GP-based path planning using Bayesian optimization

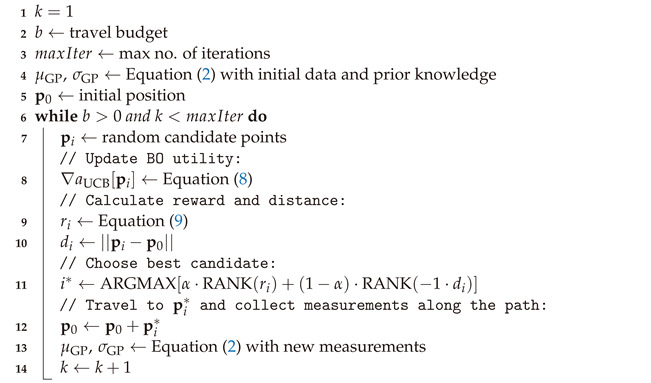



### 5.1. IPP Simulation Setup

The performance of the proposed IPP approach is evaluated for the physics-informed mean functions presented in Section 3.1. The 3GPP-based mean functions ($m_{3\mathrm{GPP}}$ and $m_{3\mathrm{GPP}-\mathrm{UMi}}$) are disregarded due to their inferior performance in the previous evaluation (cf., Section 4.1). Fixed travel budgets of 250 m and 500 m are compared. For each, a maximum of 25 iterations is set. In each iteration, 20×20×5=2000 candidate points $p_{i}$ in the latitude, longitude, and altitude directions, respectively, are drawn at random. Different configurations for the tuning parameters (κ and α) are used for each mean function. Each simulated run uses the same start location, with no training points influentially in the middle of the cluster shown in Figure 5d. The UAV simulated by AirSim travels at a fixed speed of v=10 m/s and collects points at 1 Hz, similar to the hardware setup. An ROS node that publishes the linear interpolated measurement based on the simulated UAV location is used to mimic the 5G measurement unit. Once either the travel budget or the maximum number of iterations is reached, again, the RMSE between the predicted value of the utilized GP and the ground-truth measurements is calculated. The search space is defined by the area where ground-truth measurements (see Section 2) are available. Due to the lack of measurement data between the two cuboids (see Figure 3), only the left cuboid is used as the ground truth and respective search space.

The IPP algorithm is also implemented in Python using the GPFlow package [16] in combination with TensorFlow [17].

### 5.2. IPP Simulation Results

The evaluation results for the different travel budgets are shown in Figure 8. The root-mean-square error (RMSE) between the measurement from the test set and the predicted value of the GP with the indicated mean function ($m_{\mathrm{ZERO}}$, $m_{\mathrm{CD}}$, $m_{\mathrm{SIM}}$) and the tuning parameters of the IPP algorithm (α and κ) is plotted using a logarithmic color scale. The maximum observed RMSE is 14.41 dBm for the distance-based mean with α=0.5 and κ=0.0.

Similar to the preceding evaluation, the simulation-based mean is mostly independent of both the travel budget and the IPP tuning parameters (α and κ). The distance-based mean and the zero mean, on the other hand, both show improvement with an increasing travel budget.

### 5.3. IPP Simulation Discussion

The zero mean achieves the lowest RMSE for both travel budgets. However, a clear trend of the travel budget or the candidate selection parameters is not connected to those individual low results. The simulation-based mean, on the other hand, enables a very consistently low RMSE.

For α=0, IPP candidate selection is independent of the GP prediction. Thus, adjusting κ is meaningless. Nevertheless, multiple repeated runs highlight the influence of the random candidate draw on the results. This shows that the performance of both the distance-based and zero mean relies on the candidate draw. In contrast to the results of the evaluation presented in Section 4, the distance-based mean cannot provide better results than the zero mean. This might be due to the antenna orientation (see Figure 3) that is not modeled by the distance-based mean.

Both tested travel budgets enable a significant reduction compared to the distance traveled in the measurement trials. The mileage for the two flights needed for the cuboid on the left in Figure 3 totals more than 13 km, resulting in a flight distance reduction of up to 98%. While also considering the halved flight speed of 5 m/s instead of 10 m/s, the reduction in flight time is limited by the hardware and implementation-dependent computation time for GP training and optimization.

## 6. Conclusions and Future Work

The proposed concept and demonstrator show the potential of using real-time surrogate GP modeling for efficient online path planning of a drone-based 5G measurement system, opening up the potential for a systematic yet efficient characterization and monitoring of 5G campus networks in all three dimensions. Based on real-world 5G-RSRP measurements in a cellular 5G-SA campus network at the Skandinavienkai of the Port of Lübeck, Germany, we studied the incorporation of prior knowledge into GP mean functions for the scenario of an aerial measurement unit tasked with efficiently mapping a cellular network site. While the results are highly dependent on the test settings, it can be said that the prediction performance can be improved through the use of an appropriate mean prior. If the relation of the training points to the future prediction locations is known, the results of this study can be used to choose the best-fitting GP mean. For prediction locations that cover the same area as the training points, the evaluation results suggest that the distance-based mean (Equation 5) is a good choice. If the prediction locations are known to be in a different area than the training points, incorporating as much prior information as possible into the mean is useful. This is the case for the mean prior based on detailed ray-tracing simulations (Equation 4). For mobile network sites, this also allows for the training the GP hyperparameters on measurements of one base station to improve the predictions for another base station with few to no measurements.

Subsequently, an IPP algorithm with a fixed travel budget using Bayesian optimization was developed and validated in a realistic simulation. The GP mean prior based on detailed ray-tracing simulations (Equation 4) can reliably reconstruct the data points from the measurement trial with an average RMSE below 6 dBm and 98% less distance traveled. The promising performance of the distance-based mean in the evaluations was not observed in the IPP simulations.

In both the mean function evaluation and the IPP simulation, the RMSE achieved by the GP models is below the RMSE of two subsequent measurement trials (see Section 2). This highlights a significant limitation of the implemented GPs, i.e., their inability to model temporal changes in signal strength. While time could be integrated as an additional input to the GP, doing so would exponentially increase the computational complexity of hyperparameter learning and the subsequent optimization problem, presenting a practical constraint.

The presented IPP algorithm stands out compared to other related approaches [10,11,12,14], as it considers a fixed travel budget, in addition to running online onboard the UAV. Differences in goals and methodology hinder a comparison of its mapping capabilities with those of our approach.

As part of a research project, multiple GPs with a simulation-based mean function are used to create a live map visualization of a mobile network site with multiple cell towers. New measurements from continuous terrestrial mobile measurement units are used to update the GP. If no recent data are available, the GP prediction naturally converges to the ray-tracing simulation.

The ongoing work also aims at testing the IPP algorithm in real-world settings. Thanks to the ROS framework, a transition to the UAV hardware can easily be achieved once required obstacle avoidance functionality is integrated into the algorithm. This could be achieved, e.g., through the integration of LiDAR or camera sensor data and subsequent restrictions on the search space for the selection of candidate points. 

## Figures and Tables

**Figure 1 sensors-24-07601-f001:**
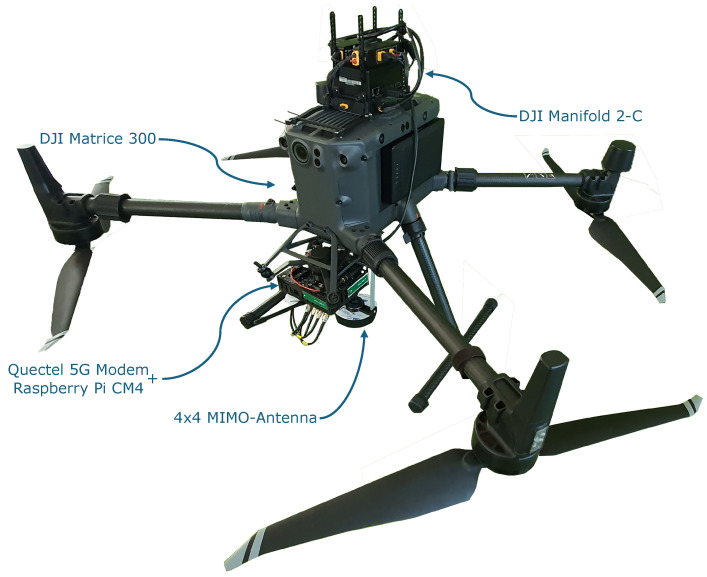
5G mMeasurement unit mounted below a DJI Matrice 300 UAV.

**Figure 2 sensors-24-07601-f002:**
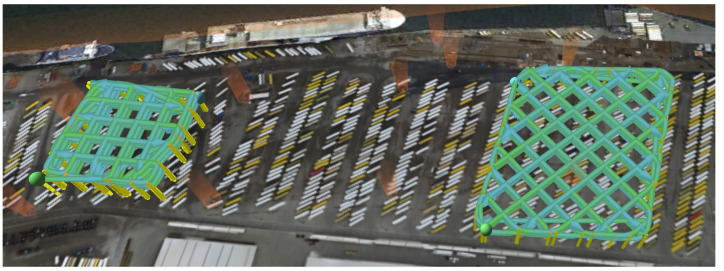
Flight planning for the measurement flight in UgCS ground control software by SPH Engineering (Riga, Latvia). The green lines depict the flight routes, and the red transparent areas mark no-fly zones around higher structures like light poles.

**Figure 3 sensors-24-07601-f003:**
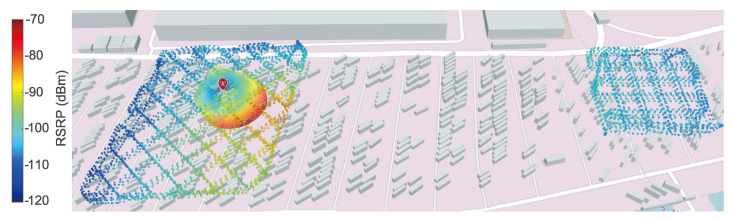
Scatter plot of the received reference signal strength (RSRP) measurements, together with the antenna pattern of the base station, on a geospatial map. The base station is marked with a drop pin. Created using the MATLAB Antenna Toolbox. Background: © OpenStreetMap contributors, CC BY-SA.

**Figure 4 sensors-24-07601-f004:**
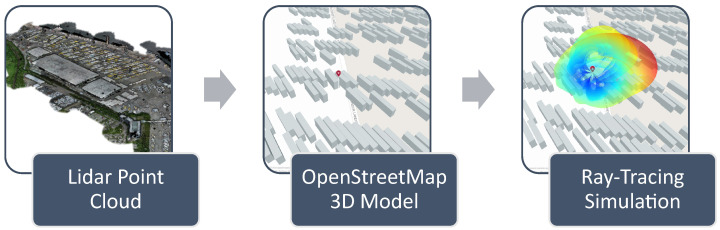
Visualization of the processes of acquiring the simulation data. Background: © OpenStreetMap contributors, CC BY-SA.

**Figure 5 sensors-24-07601-f005:**
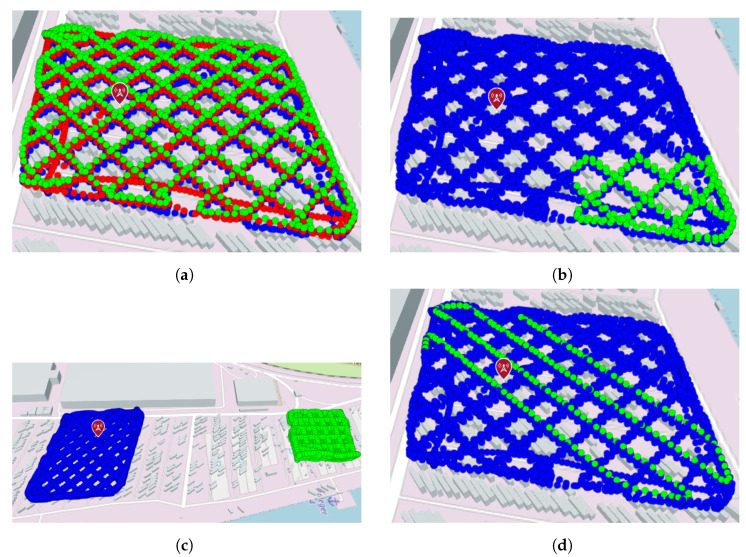
Evaluation setups with different means to separate the test set from the training set. Training point (candidates) are labeled in green, test points are labeled in blue, and discarded points are labeled in red. The base station is marked with a drop pin. (**a**) Height separation. (**b**) Cluster separation. (**c**) Distance separation. (**d**) Random split (example). Background: © OpenStreetMap contributors, CC BY-SA.

**Figure 6 sensors-24-07601-f006:**
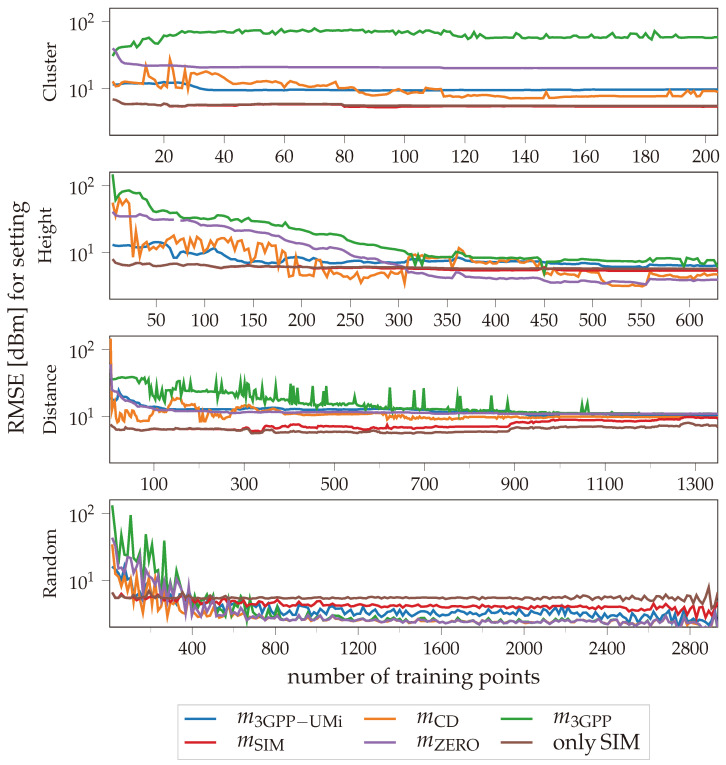
Root-mean-square error (RMSE) between the measurement from the test set and the predicted value of the utilized GP with the indicated mean function ($m_{3\mathrm{GPP}-\mathrm{UMi}}$, $m_{\mathrm{CD}}$, $m_{3\mathrm{GPP}}$, $m_{\mathrm{SIM}}$, $m_{\mathrm{ZERO}}$) or the simulation without a GP (“onlySIM”) on a logarithmic axis.

**Figure 7 sensors-24-07601-f007:**
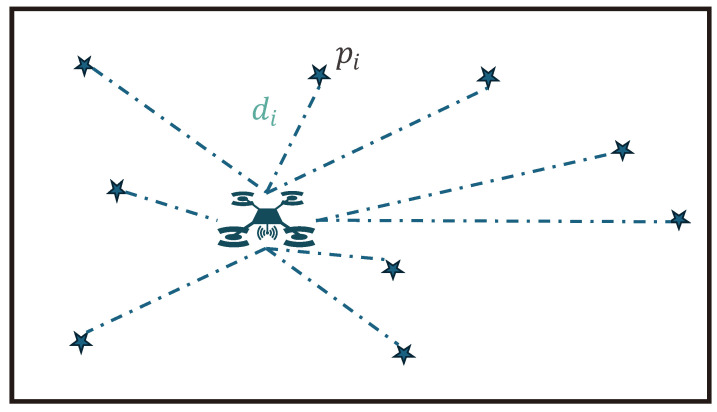
Simplified 2D visualization of the candidate points.

**Figure 8 sensors-24-07601-f008:**
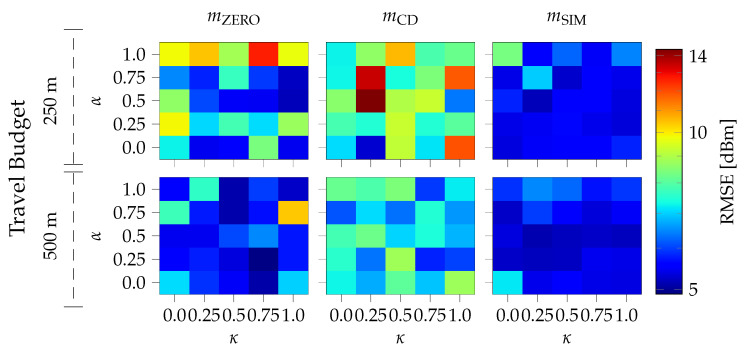
Root-mean-square error (RMSE) between the measurement from the test set and the predicted value of the utilized GP with the indicated mean function ($m_{\mathrm{ZERO}}$, $m_{\mathrm{CD}}$, $m_{\mathrm{SIM}}$) and the tuning parameters of the IPP algorithm (α and κ) plotted using a logarithmic color scale.

## Data Availability

The data presented in this research are available on request from the corresponding author. The data are not publicly available due to technical reasons.

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
