# Peer review of "Informative Path Planning Using Physics-Informed Gaussian Processes for Aerial Mapping of 5G Networks"

_sensors, 2024, doi:10.3390/s24237601_

Round 1
Reviewer 1 Report
Comments and Suggestions for Authors
This paper is based on the real-world measurement data of a 5G-SA campus network in Lübeckport, Germany, which provides a solid basis for model evaluation as real-world conditions on the ground. Experimental results show that the prediction accuracy of Gaussian process model can be significantly improved by using appropriate mean functions, and this point is also highlighted in the information path planning simulation. These findings have practical application value for UAV path planning and 5G network deployment.
However, I think there is room for improvement in the following areas:
1. Did the author try other datasets of other real-world measurements, or other scenarios to test his prediction model?
2. Invite the author to add a discussion of the limitations of the model and possible future improvements in the discussion section.
3. Please check and make sure that all charts and formulas are clear and readable in the final version.
4. It is suggested that the authors further emphasize the practical application value and potential impact of the study on 5G network deployment in the conclusion.
5.Depth of data analysis: Although the paper mentions the use of Gaussian process model for signal strength prediction, the selection of model parameters and the optimization process are not described in detail enough. It is suggested that the authors can further elaborate on the basis for determining the model parameters and the impact of different parameter settings on the model performance.
6.Comparative analysis of experimental results: Although some experimental results are shown in this paper, there is a lack of comparative analysis with other methods or algorithms. It is suggested that the authors can add some comparative experiments to highlight the advantages and limitations of the proposed method.
Author Response
We would like to thank the reviewer for the valuable comments, which helped us to improve the manuscript. Please find a detailed reply to the reviewer's comments below.
Comments 1: Did the author try other datasets of other real-world measurements, or other scenarios to test his prediction model?
Response 1: We fully agree that the analysis on more datasets would indeed support the validity of the prediction model. However, to the best of our knowledge, no applicable datasets are available which would allow testing the combination of the path planning algorithm with the physics-informed Gaussian process. Other datasets lack supplementary information, like ray tracing data or locations of the base stations, to name just two. It was thus not possible to perform such a comparison. As outlined in the "Conclusion and Future Work" section, our ongoing, and future work aims at testing the IPP algorithm in several different real-world settings.
Comments 2: Invite the author to add a discussion of the limitations of the model and possible future improvements in the discussion section.
Response 2: We thank the reviewer for this important remark. One of the main limitations of the implemented GPs is their inability to model temporal changes. This issue is highlighted in the “Conclusion & Outlook” section.
Comments 3: Please check and make sure that all charts and formulas are clear and readable in the final version.
Response 3: We thank the reviewer for this remark. We have carefully checked all charts and formulas for the resubmission and fixed the captions in Figure 6.
Comments 4: It is suggested that the authors further emphasize the practical application value and potential impact of the study on 5G network deployment in the conclusion.
Response 4: We thank the reviewer for this valuable remark: We have added the following sentence to the “Conclusion section”: “The proposed concept and demonstrator show the potential of using real-time surrogate GP modeling for efficient online path-planning of a drone-based 5G measurement system, opening up the potential for a systematic yet efficient characterization and monitoring of 5G campus networks in all three dimensions. “
Comments 5: Depth of data analysis: Although the paper mentions the use of Gaussian process model for signal strength prediction, the selection of model parameters and the optimization process are not described in detail enough. It is suggested that the authors can further elaborate on the basis for determining the model parameters and the impact of different parameter settings on the model performance.
Response 5: We thank the reviewer for this comment. The optimization process is now described in the pseudocode. However, as the scope was on the general applicability of the GP-surrogate modeling for path planning, we focused on the evaluation of the GP prior as this was identified as a central element of this approach. The hyperparameters of the GP, i.e., the length scales and variances of the kernel as well as parameters of the mean function, have been optimized based on the measurement data. A full assessment of the impact of every single GP parameter and function on the modeling and path planning performance was beyond the scope of this article.
Comments 6: Comparative analysis of experimental results: Although some experimental results are shown in this paper, there is a lack of comparative analysis with other methods or algorithms. It is suggested that the authors can add some comparative experiments to highlight the advantages and limitations of the proposed method.
Response 6: We thank the reviewer for this comment. The aim of this article was the introduction and proof of general applicability of the proposed modeling and path planning concept for a drone-based 5G measurement system. To the best of our knowledge, this was the first time, that such a path-planning approach based on a GP surrogate model has been developed for an aerial 5G measurement system. We agree with the reviewer that further test flights are needed in order to fully characterize the performance and potential improvements of the proposed approach. We tried to emphasize this in the outlook of the article.
Reviewer 2 Report
Comments and Suggestions for Authors
1. Authors should clearly specify their scientific contribution.
2. The paper would be more valuable if pseudocode is added (for the algorithm that combines the physics-informed GP models with Bayesian optimization)
3. What happens when there are obstacles in the area of interest (in Figure 2 the obstacles are outside the flight zone)? What if the wind is blowing?
4. If there is a comparison with other similar (existing and tested) methods, why are the comparisons not listed in a table?
Author Response
We would like to thank the reviewer for the valuable comments, which helped us to improve the manuscript. Please find a detailed reply to the reviewer's comments below.
Comments 1: Authors should clearly specify their scientific contribution.
Response 1: We are thankful for this comment. The three main contributions of the article are:
1) The first main contribution of this article is the extension of the idea of Di Taranto et al. 2014 by analyzing the influence of different physics-informed mean priors of varying complexity, including a full ray tracing simulation, as GP mean function.
2) The second main contribution is the application of the GP-model with physics-informed mean functions as a basis for informative path planning (IPP) of the UAV flight path.
3) The development of a UAV-based 5G measurement system that is capable of efficiently performing a 3-dimensional mapping of a 5G network site within the premises of the Port of Lübeck using physics-informed GP-based data processing
We have restructured the introduction to emphasize these points more clearly.
Comments 2: The paper would be more valuable if pseudocode is added (for the algorithm that combines the physics-informed GP models with Bayesian optimization)
Response 2: We agree that pseudocode would indeed contribute to the understanding of the article. We added this as “Algorithm 1” in Section 5. Thank you for this suggestion.
Comments 3: What happens when there are obstacles in the area of interest (in Figure 2 the obstacles are outside the flight zone)? What if the wind is blowing?
Response 3: We are thankful for this question. The obstacle avoidance functionality, albeit part of the safety features of the chosen drone platform, is an ongoing endeavor. It can, for instance, be integrated by using sensor data (like LiDAR) and limiting the search space for selecting the subsequent candidate point. We tried to clarify a corresponding sentence in the last section of the conclusion.
Comments 4: If there is a comparison with other similar (existing and tested) methods, why are the comparisons not listed in a table?
Response 4: To the best of our knowledge, this was the first time, that such a path-planning approach based on a GP surrogate model has been developed for an aerial 5G measurement system.